# Plant Growth Promotion and Stress Tolerance Enhancement through Inoculation with *Bacillus proteolyticus* OSUB18

**DOI:** 10.3390/biology12121495

**Published:** 2023-12-06

**Authors:** Piao Yang, Wenshan Liu, Pu Yuan, Zhenzhen Zhao, Chunquan Zhang, Stephen Obol Opiyo, Ashna Adhikari, Lijing Zhao, Garrett Harsh, Ye Xia

**Affiliations:** 1Department of Plant Pathology, College of Food, Agricultural and Environmental Sciences, The Ohio State University, Columbus, OH 43210, USA; yang.4636@buckeyemail.osu.edu (P.Y.); wzl0028@auburn.edu (W.L.); yuan.1282@osu.edu (P.Y.); zzhao9@wpi.edu (Z.Z.); opiyo.1@osu.edu (S.O.O.); adhikari.168@buckeyemail.osu.edu (A.A.); ljzhao@ucdavis.edu (L.Z.); harsh.68@buckeyemail.osu.edu (G.H.); 2College of Agriculture and Applied Sciences, Alcorn State University, 1000 ASU Dr. #690, Lorman, MS 39096, USA; czhang@alcorn.edu

**Keywords:** beneficial microbes, *Bacillus proteolyticus*, *Botrytis cinerea*, *Pseudomonas syringae*, *Arabidopsis thaliana*, induced systemic resistance (ISR), systemic acquired resistance (SAR), plant immunity, transcriptome analysis, Metascape analysis

## Abstract

**Simple Summary:**

*Bacillus proteolyticus* OSUB18, isolated from switchgrass, has previously been identified for its role in enhancing plant growth and suppressing diseases. In this study, its effect on the model plant, *Arabidopsis thaliana*, is explored. OSUB18 was found to promote the growth and health of *Arabidopsis* under various stress conditions. Advanced RNA-seq technology revealed changes in gene expression in plants treated with OSUB18, with the results aligning with phenotypic and metabolic observations. OSUB18 also primes *Arabidopsis* to effectively resist different phytopathogens and showcases strong antagonistic effects against various pathogens in vitro. In conclusion, OSUB18 boosts plant growth and health by altering root architecture, defense signaling, and reducing biotic and abiotic stresses.

**Abstract:**

The isolation of *B. proteolyticus* OSUB18 from switchgrass unveiled its significant potential in both the enhancement of plant growth and the suppression of plant diseases in our previous study. The elucidation of the related mechanisms governing this intricate plant–microbe interaction involved the utilization of the model plant *Arabidopsis thaliana*. In our comprehensive study on *Arabidopsis*, OSUB18 treatment was found to significantly alter root architecture and enhance plant growth under various abiotic stresses. An RNA-seq analysis revealed that OSUB18 modified gene expression, notably upregulating the genes involved in glucosinolate biosynthesis and plant defense, while downregulating those related to flavonoid biosynthesis and wound response. Importantly, OSUB18 also induces systemic resistance in *Arabidopsis* against a spectrum of bacterial and fungal pathogens and exhibits antagonistic effects on phytopathogenic bacteria, fungi, and oomycetes, highlighting its potential as a beneficial agent in plant stress management and pathogen resistance. Overall, our findings substantiate that OSUB18 exerts a stimulatory influence on plant growth and health, potentially attributed to the remodeling of root architecture, defense signaling, and the comprehensive mitigation of various biotic and abiotic stresses.

## 1. Introduction

Beneficial microorganisms encompass a diverse array of microbial entities that confer multifaceted advantages to plants, animals, and human endeavors. Their manifold contributions include facilitating nutrient provisioning, invigorating plant growth, orchestrating biocontrol measures against phytopathogens, ameliorating soil structure, orchestrating the bioaccumulation of inorganic compounds, and undertaking bioremediation activities in metal-laden soils [1]. In the realm of botanical ecosystems, beneficial microorganisms manifest as entities capable of establishing residence within plant roots or other botanical structures, thereby elevating growth trajectories, bolstering health indices, and augmenting productivity through a spectrum of nuanced mechanisms [2]. The elucidation of their pivotal roles in nurturing plant growth and fortifying host defenses against biotic and abiotic stresses has emerged as a focal point for researchers and practitioners concerned with the domains of agriculture and environmental science. Unveiling the intricacies governing the synergistic interplay between beneficial microorganisms and their host plants holds profound significance. This pursuit underscores the cultivation of sustainable, environmentally harmonious agricultural practices that proffer augmented crop yields and elevated product quality, concomitant with a curtailed reliance on chemical fertilizers and harmful pesticides. Furthermore, this vantage point imparts crucial insights into the dynamics of plant–microbe interactions across diverse environmental contexts, bearing implications for the modulation of ecosystem functionalities and services at large.

*Bacillus proteolyticus*, classified within the genus *Bacillus*, is a bacterial species originally discovered in sediment derived from the Pacific Ocean. This microorganism has shown indications of possessing probiotic attributes, as noted in a recent study, where Zeng et al. [3] evaluated the probiotic potential of *B. proteolyticus* isolates (named Z1 and Z2) with antioxidant activity tests. Our prior investigations centered around the influence exerted by a distinctive bacterial variant, *B. proteolyticus* OSUB18, upon the immune response of *Arabidopsis thaliana*, a model plant. Specifically, OSUB18 was found to trigger induce systemic resistance (ISR) by heightening callose deposition, facilitating reactive oxygen species (ROS) generation, amplifying the presence of specific phytohormones and metabolites intrinsic to plant defense, and upregulating the expression of multiple defense-related genes, as detailed in our previous study [4]. However, while these insights offer a glimpse into OSUB18’s engagement with plant immunity, its comprehensive roles in fostering plant growth through the modulation of root development and host gene expression in different conditions remains enigmatic. Furthermore, whether OSUB18’s inhibitory effects extend to a broader array of phytopathogens beyond *Pseudomonas syringae* and *Botrytis cinerea* remains an important question to be answered. Further insights into the interplay between OSUB18 and plant root development, coupled with the modulation of host plant gene expression, can serve to substantiate its potential as an agent of plant growth promotion. This, in turn, can pave the way for more refined strategies for harnessing the benefits of OSUB18 in the realm of agricultural applications.

The primary objective of this study is to elucidate the multifaceted roles assumed by OSUB18 in facilitating plant growth promotion and regulating host plant gene dynamics potentially involved in stress tolerance against biotic and abiotic stresses. The overarching hypothesis posits that OSUB18 may orchestrate a recalibration of the host plant’s root development, thereby fostering the robust expansion of above-ground structures and enhancing the overall yield. Additionally, it is conjectured that OSUB18 might serve as a mitigative agent against an assortment of abiotic stresses encountered by host plants during their growth cycle. In pursuit of these goals, *Arabidopsis* Col-0 plants were cultivated on agar plates and subsequently subjected to OSUB18 inoculation, facilitating a comprehensive analysis of the consequential root development patterns exhibited by the host plants. Furthermore, to ascertain OSUB18’s potential in abiotic stress alleviation, *Arabidopsis* Col-0 plants grown in a soil substrate were strategically exposed to varying abiotic stress conditions. This enabled the discernment of whether OSUB18 pretreatment engendered a protective effect against these adverse conditions, thus bolstering the resilience of the host plants. Moreover, the investigative process encompassed the extraction of total RNA from wild-type *Arabidopsis* Col-0 plants that underwent distinct treatments, namely, water-drenching versus OSUB18-drenching treatments. This extracted RNA underwent a rigorous RNA-seq analysis, facilitating an intricate exploration of the nuanced alterations in the host plant’s gene expression landscape. Through these meticulously designed methodologies, this study endeavors to unravel the intricate interplay between OSUB18 and its host plants, thereby contributing to a more holistic comprehension of the mechanisms underpinning plant growth promotion and abiotic stress mitigation.

Our investigation illuminated the profound influence of OSUB18 on root architecture modulation, consequently fostering robust plant growth even when confronted with an array of diverse abiotic stressors in *Arabidopsis* Col-0 plants. The elucidation of this intricate relationship is substantiated by RNA-seq and a Metascape analysis [5], which unveiled both qualitative and quantitative differentials in the gene expression of *Arabidopsis* Col-0 plants, stemming from the drenching application of OSUB18. Furthermore, OSUB18 was bee unveiled as a booster for the induction of systemic resistance in host plants, effectively bolstering their defense mechanisms against both bacterial and fungal phytopathogens in a live setting. In controlled in vitro environments, OSUB18 proved to be an efficacious adversary, countering phytopathogenic bacteria [4], fungi, and oomycetes. Our study not only bridges the crucial gaps in comprehending the potential of *B. proteolyticus* to serve as a growth promoter and phytopathogen suppressor, but also underscores the dynamic mechanisms by which it achieves these effects, encompassing the recalibration of root development, nuanced alterations in host plant gene expression, the activation of systemic resistance against pathogen infections, and the enhanced tolerance against abiotic stresses.

## 2. Materials and Methods

### 2.1. Plant Materials and Growth Conditions

*Arabidopsis* plants were grown on 1/2-strength Murashige and Skoog (MS) plates or soil during short days (8 h light/16h dark cycle) at 22 °C ± 3 °C [6]. The seeds were in the wild-type Col-0 background and obtained from the *Arabidopsis* Biological Resource Center (ABRC) [7]. *Arabidopsis* seeds were surface sterilized in 50% bleach with 0.05% Tween 20 detergent for 10 min and rinsed 7 times with sterile distilled water. After storing them at 4 °C for 3 days, we sowed the sterile seeds on 1/2-strength MS plates or soil. Tomato plants (cultivar Heinz) were grown in soil in a greenhouse during long days (16 h light/8 h dark cycle) at 25 °C ± 5 °C [8,9]. The soil utilized in our experiments was Lambert LM-111 potting mix. For growing *Arabidopsis*, we used pots measuring 3.5 inches in length and width, and 2 inches in height. In contrast, tomato plants were cultivated in larger containers, specifically 2-gallon pots acquired from the Greenhouse Megastore. To ensure an adequate nutrient supply, we used Osmocote’s balanced fertilizer with an N–P–K ratio of 14–14–14. To maintain consistency and reliability in our data, each experimental iteration involved the cultivation of a minimum of six plants per condition.

### 2.2. Plant Growth Analysis

Sterile *Arabidopsis* Col-0 seeds were sown onto 10 cm Petri dishes containing 25 mL of agar-solidified 0.5× Murashige and Skoog (MS) medium. Following 3 days of vernalization at 4 °C, 10 µL droplets of a bacterial suspension with an optical density of 600 nm (OD600) at 0.1 were administered 5 cm away from the seeds. Subsequently, the inoculated Petri dishes were placed vertically in a plant growth chamber. The chamber was set at 22 °C with a 12 h light/dark cycle to promote seed germination and plant growth. The fresh and dry weights of *Arabidopsis* plants were measured on an analytical balance as previously described [4]. The *Arabidopsis* seedlings were pictured with a digital camera (Nikon D5200, Nikon, Tokyo, Japan). The primary roots, lateral roots, and root hairs were pictured with a macroscope (Olympus Macro View MVX10, Olympus Corporation, Tokyo, Japan). ImageJ software (Version 1.52t 30 January 2020) was used to measure the root length [10]. An assessment of the abiotic stress response of *Arabidopsis* plants after OSUB18 treatment was performed as follows: *Arabidopsis* seedlings were drenched with OSUB18 (10^7^ CFU/mL) or water (Ctrl) 3 times and then exposed to drought stress (by withholding irrigation), salty stress (by irrigation with 250 mM of NaCl), or cold stress (by growing them in a 4 °C cold room) for 1 week before the corresponding plant biomasses were measured. For the drought stress assay, plants treated with OSUB18 or water were subjected to drought conditions by withholding irrigation for one week. This was followed by a recovery phase where the plants were re-watered for three days. The effectiveness of the treatments was assessed by measuring the fresh weight of the plant shoots post-recovery. In the case of the salt stress assay, *Arabidopsis* plants, after being treated with OSUB18 or water, were exposed to saline conditions. This was achieved by drenching the plants with a 250 mM NaCl solution, 1 week prior to the fresh weight measurement of the plant shoots. For the cold stress assay, the experimental setup involved growing *Arabidopsis* plants treated with either OSUB18 or water at a lower temperature. Instead of the standard 22 °C, these plants were cultivated at 4 °C. Subsequently, the impact of this cold treatment was evaluated by measuring the fresh weight of the plant shoots. These methodologies were carefully designed to assess the impact of OSUB18 treatment on *Arabidopsis* plants under different environmental stress conditions, providing valuable insights into the plant’s stress response mechanisms.

### 2.3. RNA-Seq and Bioinformatics Analysis

*Arabidopsis* plants, after three drenching treatments (OSUB18 vs. water), were collected for total RNA extraction using TRIzol reagent (Invitrogen, Waltham, MA, USA) following the manufacturer’s instructions. The total RNA samples were treated with DNase I to remove potential DNA residues. DNA-free RNA samples were used for library preparation and RNA-sequencing. RNA-seq was conducted on an Illumina HiSeq4000 PE150 platform by the company Novogene Bioinformatics Technology Co., Ltd. (Beijing, China). FastQC software was used to check the quality of the reads (https://www.bioinformatics.babraham.ac.uk/projects/fastqc/, accessed on 1 January 2019). The Cutadapt package was used to remove sequencing adaptors [11]. The FASTX-Toolkit was used to discard reads shorter than 30 bp (http://hannonlab.cshl.edu/fastx_toolkit/index.html, accessed on 1 January 2019). The SortMeRNA toolkit was used for in silico rRNA depletion [12]. Trimmomatic software (version 0.40) was used to remove low-quality reads [13]. The processed reads were then mapped against the *Arabidopsis* genome Araport11 [14] with the HISAT2 package [15]. We used SAMtools to convert sorted mapping files into a BAM format according to the previous report [16]. The related sequencing data were deposited in the NCBI database under the BioProject accession number PRJNA1013281. StringTie [17] was used to infer expressed genes and transcripts. The DESeq2 package [18] was used for the differential gene expression analysis according to the StringTie manual [17]. The *p*-values were adjusted by using the Benjamini–Hochberg method [19]. Genes with an adjusted *p*-value < 0.05 and log2 fold change >1 were assigned as differentially expressed. The web-based tool (http://bioinformatics.psb.ugent.be/cgi-bin/liste/Venn/calculate_venn.htpl, accessed on 8 January 2019) was used to compare the genes identified in the RNA-seq data, and gene ontology (GO) analyses were performed with Metascape [5]. We also used Mapman software (Version 3.6.0RC1) [20] for annotation and enrichment information. The heatmaps were constructed using a subset of the RNA-seq data with the webserver OmicShare (https://www.omicshare.com/tools/, accessed on 15 January 2019). Gene descriptions were retrieved from the TAIR database [21].

### 2.4. Pathogen Inoculations, Phloem Exudate Collection, and SAR Assay

A bacterial pathogen infection was conducted according to the previous study [22] as follows: 4~6-week-old plant leaves were syringe injected with *Pseudomonas syringae* pv. tomato DC3000 (*Pst* DC3000) at 1 × 10^6^ CFU/mL or dipping inoculated with *Pst* DC3000 at 5 × 10^8^ CFU/mL containing 0.05% Silwet L-77. A total of 2–3 days later, the bacterial virulence was quantified by measuring *Pst* DC3000 growth in infected leaves using the serial dilution technique according to the previous study [23]. A fungal pathogen infection was conducted according to [4] as follows: 4~6-week-old plant leaves were inoculated with a single droplet of *B. cinerea* conidia at 5 × 10^5^ conidia/mL, and infected plants were maintained with high humidity levels to facilitate disease development. A total of 2–3 days later, disease symptoms were assessed by measuring the lesion size using ImageJ software [10]. The phloem exudate (PEX) collection of *Arabidopsis* was performed according to the previous study [24] after the leaves were infiltrated with the indicated bacteria OSUB18 (5 × 10^7^ CFU/mL) for 24 h. An SAR assay was conducted according to the previous studies with minor modifications [25,26,27]. In this experiment, local leaves of the plant were initially treated with beneficial bacteria (like OSUB18) at a concentration of 1 × 10^6^ CFU/mL or with a solution containing substances secreted by these bacteria, known as cell-free PEX. Two days after this initial treatment, leaves that were not treated (distant leaves) were infected with plant pathogens, either *Pst* DC3000 or *B. cinerea* spores. Following an additional period of 2–3 days, we observed and evaluated the disease symptoms. For the SAR assay, *Pseudomonas syringae* pv. tomato strain DC3000 (*Pst* DC3000) expressing the avirulent effector avrRpt2 (avrRpt2 in short) was used as the positive control for SAR induction (SAR+) via leaf infiltration at 1 × 10^6^ CFU/mL. Water was used as the negative control for SAR induction (SAR−). SAR is generally activated when a plant detects an attack by a pathogen at the site of infection. This detection often involves plant receptors that identify effector proteins derived from plant pathogens (such as the avrRpt2 effector protein) [28]. Additionally, recent research has shown that certain chemical molecules, such as extracellular pyridine nucleotides [25] and N-hydroxypipecolic acid [26], can effectively induce SAR in *Arabidopsis*.

### 2.5. Microbial Materials and Antagonistic Assays

All the bacteria were stored in 25% glycerol at −80 °C for long-term storage. Beneficial bacteria, such as OSUB18 and *Pseudomonas fluorescens* Pf5, were grown on tryptic soy agar (TSA) at 28 °C for short-term use. OSUB18 is a bacterial strain that has garnered attention for its capacity to bolster resistance in *Arabidopsis* plants against both bacterial and fungal pathogens [4]. Conversely, *P. fluorescens* Pf5 serves as a commensal bacterium residing in the rhizosphere, where it produces secondary metabolites that inhibit soilborne pathogens affecting plants [29]. Phytopathogenic *P. syringae* bacteria were grown on King’s B agar (KBA) containing 50 mg/L of rifampicin (for *Pst* DC3000) and/or 50 mg/L of kanamycin (for avrRpt2). Fungal and oomycete phytopathogens were grown on potato dextrose agar (PDA) at 20–28 °C for short-term use. Antagonistic assays of OSUB18 against different phytopathogens were conducted on agar plates in vitro at RT as modified from the previous study [4].

### 2.6. Statistical Analysis

All statistical analyses were performed using GraphPad software (v9.0.2). The data were presented as box and whisker plots (minimum to maximum show all points). The exact *p*-values (two-tailed unpaired *t* test or Tukey’s multiple comparisons test) are shown in each graph. The related experiments were repeated three times. The representative data from three biological replicates are presented.

## 3. Results

### 3.1. Alterations in Root Architecture and the Promotion of Plant Growth under Various Abiotic Stresses in Arabidopsis Are Discerned through the Impact of OSUB18

The aforementioned findings underscore the substantive role of OSUB18-drenching treatment in significantly enhancing the growth of *Arabidopsis* Col-0 plants, as expounded in the study [4]. However, insights into OSUB18’s contributions to plant root development and the related functions have not been well studied. Leveraging the premise of OSUB18’s capacity to yield beneficial metabolites, such as ammonia and indole-3-acetic acid (IAA) [4], we postulated a potential transformation in the root architecture of *Arabidopsis* Col-0 plants following OSUB18 treatment. To empirically validate this hypothesis, surface-sterilized Col-0 seeds were sown on ½ MS media and subsequently inoculated with either sterile water (Ctrl) or fresh OSUB18 cells, oriented in a divergent direction relative to the agar plate. This experimental setup afforded an enhanced visualization of *Arabidopsis* Col-0 root development. Following a span of two weeks, consistent with our antecedent findings for soil pots [4], a conspicuous enhancement in the growth of *Arabidopsis* Col-0 seedlings was observed upon OSUB18 inoculation on ½ MS media (Figure 1A). In other words, while a reduction in the primary root length was evidenced (Figure 1D), OSUB18-treated *Arabidopsis* Col-0 plants exhibited the development of enlarged primary roots (Figure 1B,C,E), increased density and number of root hairs (Figure 1B,C,F) characterized by extended lengths (Figure 1B,C,G), and a greater propensity for lateral root formation (Figure 1A,H), leading to a significant increase in the seedling fresh weight (Figure 1A,I).

Significantly, the OSUB18-drenching treatment not only demonstrated the potential for augmenting standard plant growth (Figure 1J,K), but also exhibited resilience in countering abiotic stressors, such as cold temperatures (Figure 1K), elevated salinity (Figure 1L), and drought conditions (Figure 1M). We proposed that the enhanced root development observed with OSUB18 treatment could potentially benefit the plant by improving nutrient uptake from the environment. This, in turn, might enhance the plant’s adaptability and resilience in the face of adverse abiotic stress conditions in the context of *Arabidopsis* Col-0 plants.

### 3.2. RNA-Seq Analysis Reveals Qualitative and Quantitative Differences in Arabidopsis Gene Expressions Due to OSUB18-Drenching Treatment

To elucidate the intricacies of OSUB18’s influence, we executed an experimental regimen involving the sampling of *Arabidopsis* Col-0 plants, cultivated in soil drenched with either water (Ctrl) or OSUB18. The resultant total RNA was meticulously extracted and subjected to a comprehensive RNA-seq analysis, thereby facilitating a comprehensive exploration of the shifts in the *Arabidopsis* transcriptome because of the OSUB18-drenching treatment. The suite of RNA-seq reads was strategically mapped to the *Arabidopsis* reference genome Araport11 [14], subsequently discerning differentially expressed genes through the employment of DESeq2 software (Release (3.18)) [18], which moderated the fold changes and dispersions for the RNA-seq data.

Following the stringent criteria (log2 fold change > 1; adjusted *p*-value < 0.05 relative to Ctrl treatment), the transcripts were meticulously selected for further examination, as presented in Appendix A. To facilitate a more nuanced visualization of these patterns, a heatmap was meticulously constructed, featuring a subset of genes differentially expressed in the *Arabidopsis* Col-0 plants. This subset, enriched with 40 genes of pivotal biological significance with the highest or lowest differential expressions, provided insights into OSUB18’s influence (Figure 2A). Of notable prominence, the cohort of upregulated genes in OSUB18-drenched *Arabidopsis* Col-0 plants was characterized by a pronounced enrichment of defense-related genes, inclusive of *PDF1.2b*, *PDF1.2*, and *PDF1.3* (Figure 2A). This observation underscores the activation of host resistance signaling pathways in *Arabidopsis* Col-0 plants in response to OSUB18 drenching, aligning seamlessly with our antecedent study [4].

Introducing the user-friendly annotation software “Mapman” [20], we further garnered insights into the intricate landscape of the RNA-seq data. This software facilitated the visualization of additional genes and biological processes, thereby uncovering significant alterations. As exemplified by Figure 2B, it was evident that *Arabidopsis* auxin-responsive genes, such as *WAG1* (log2 fold change = 1.49), *SAUR4* (log2 fold change = 2.13), and *SAUR8* (log2 fold change = 1.61), exhibited significant inductions following OSUB18-drenching treatment, mirroring our prior findings [4]. In concordance with our observations regarding OSUB18’s role as a potential mediator of plant abiotic stress responses (Figure 1I–M), *Arabidopsis* abiotic stress-responsive genes *MTHSC70-2* (log2 fold change = 1.41) and *CAP* (log2 fold change = 1.26) displayed pronounced upregulations (Appendix A). Intriguingly, a suite of *Arabidopsis* secondary metabolism sulfur-containing glucosinolate genes, including *BCAT4* (log2 fold change = 1.77), *IPMI1* (log2 fold change = 2.31), *CYP79F2* (log2 fold change = 3.36), and *SOT18* (log2 fold change = 1.19), was markedly upregulated. In parallel, an array of *Arabidopsis* secondary metabolism flavonoid genes, exemplified by *JOX4* (log2 fold change = −2.91), *LDOX* (log2 fold change = −5.14), *AT5MAT* (log2 fold change = −3.91), *CHIL* (log2 fold change = −2.43), *TT5* (log2 fold change = −1.90), *UF3GT* (log2 fold change = −3.92), *DFR* (log2 fold change = −6.23), and *TT7* (log2 fold change = −3.31), manifested marked downregulations (Appendix A). This dynamic cascade of gene expression changes denotes a potential mode of action for OSUB18 drenching, where it engenders the upregulation of defensive glucosinolate genes and orchestrates the downregulation of flavonoid genes, thus potentially harnessing these pathways to establish the initial interaction with *Arabidopsis* Col-0 plants.

### 3.3. OSUB18 Elicits the Upregulation of Glucosinolate Biosynthesis and Defense-Related Genes in Arabidopsis

To achieve a comprehensive molecular understanding of *Arabidopsis* plant responses following OSUB18-drenching treatment, we executed an extensive gene list analysis employing a rigorously established protocol [5], with a primary focus on the repertoire of upregulated genes (Appendix A). Substantiating this investigative approach, we identified a trio of pivotal biological processes in *Arabidopsis* that experienced a pronounced induction due to OSUB18-drenching treatment. Specifically, these included WP4597: glucosinolate biosynthesis from methionine, GO:0009908: flower development, and GO:0016114: terpenoid biosynthetic process (Figure 3A–D). Glucosinolates, as natural secondary metabolites, assume a pivotal role in plants, boasting both nutritional and defensive attributes. This cadre of components is susceptible to enzymatic hydrolysis by microbiota, while concurrently serving as indispensable contributors to plant defense mechanisms [30]. In this context, OSUB18 may incite *Arabidopsis* plants to synthesize glucosinolates as a dual-purpose asset: a nutritional resource for a mutualistic interaction and an amplified defense mechanism against potential phytopathogenic incursions. Better flower development can facilitate sexual reproduction and the production of seeds [31]. Additionally, terpenoid biosynthesis is an important metabolic process in plants, playing a critical role in their growth, development, and interaction with the environment [32].

Remarkably, an analysis of protein–protein interaction (PPI) networks gleaned from the gene list substantiated these molecular insights. Noteworthy enrichments were discerned in vital plant defense-related genes, most notably plant defensin 1.2 (*PDF1.2*, *AT5G44420*), plant defensin 1.2b (*PDF1.2b*, *AT2G26020*), and plant defensin 1.3 (*PDF1.3*, *AT2G26010*) (Figure 3E), which were ethylene- and jasmonate-responsive, and commonly used as markers for characterizing the jasmonate-dependent defense responses of plants [33]. This robust alignment accentuated the orchestrated activation of host resistance mechanisms upon OSUB18-drenching treatment. Furthermore, the PPI analysis sheds light on an interaction between nitrate transporter 3.1 (*NRT3.1*, *AT5G50200*) and nitrate transporter 2.1 (*NRT2:1*, *AT1G08090*) (Figure 3E), a phenomenon congruent with ammonia production resulting from OSUB18 activity [4]. Notably, our analysis also spotlighted a potential interaction between mitochondrial hso70-2 (*MTHSC70-2*, *AT5G09590*) and maternal effect embryo arrest 3 (*MEE3*, *AT2G21650*), an MYB transcription factor integral to the early morphogenesis of *Arabidopsis* plants (Figure 3E). These collective data underscore the premise that OSUB18-drenching treatment may precipitate a cascade of molecular events, augmenting nutrient acquisition and priming defense gene expression, effectively fortifying *Arabidopsis* against potential biotic and abiotic stresses.

### 3.4. OSUB18 Exerts the Downregulation of Plant Flavonoid Biosynthetic and Wound Responsive Genes in Arabidopsis

In pursuit of a comprehensive grasp of the nuanced influence of OSUB18, we embarked on an investigative trajectory to ascertain the plant genes impacted by OSUB18-drenching treatment. Employing a congruent gene list analysis strategy [5], our focus pivoted exclusively to the cadre of repressed genes (Appendix A). This analytical endeavor revealed a prominent substratum of biological processes in *Arabidopsis* that experienced a pronounced suppression consequent to OSUB18-drenching treatment. Specifically, these processes encompassed GO:0009813: flavonoid biosynthetic process and GO:0009611: response to wounding (Figure 4A–D). The manifold roles enacted by flavonoids, including the regulation of plant development, pigmentation, and active involvement in host–microbe interactions and other defense mechanisms have been meticulously elucidated in the scientific literature [34]. In the context of OSUB18, it is plausible that the downregulation of flavonoid synthesis by the plant may serve to augment the stable colonization of OSUB18 in *Arabidopsis* Col-0 plants. This strategic modulation of flavonoid synthesis could potentially confer an advantageous milieu for OSUB18’s establishment.

Furthermore, the observed downregulation of plant wound responsive genes after OSUB18-drenching treatment aligned seamlessly with the prevailing knowledge, positing that beneficial microbes transiently may attenuate local plant immune responses to facilitate their initial successful establishment in the host plants [35]. This intriguing alignment underscores OSUB18’s strategic adaptation to foster its harmonious interaction with *Arabidopsis* Col-0 plants. Indeed, our exploration of protein–protein interaction (PPI) networks unearthed significant interactions involving pivotal elements, such as chalcone flavanone isomerase (*CFI, AT3G55120*), cytochrome P450 75B1 (*CYP75B1*, *AT5G07990*), dihydroflavonol 4-reductase (*DFR*, *AT5G42800*), chalcone isomerase, like (*CHIL*, *AT5G05270*), and leucoanthocyanidin dioxygenase (*LDOX*, *AT4G22880*) (Figure 4E). These data illuminate the nuanced orchestration of OSUB18’s influence, selectively silencing genes in the flavonoid biosynthetic and wound responsive pathways. This astute regulatory strategy was likely instrumental in priming the conditions requisite for OSUB18’s successful initial establishment in *Arabidopsis* Col-0 plants.

### 3.5. OSUB18 Elicits Systemic Resistance in Host Plants against Bacterial and Fungal Pathogens

Our preceding RNA-seq analysis (Figure 2) afforded a compelling insight into the upregulation of plant defense genes consequent to OSUB18-drenching treatment. Of particular significance was the discernment of systemic acquired resistance (SAR), which engendered resistance in remote tissues subsequent to local immunization by necrotizing pathogens, such as *Pst* DC3000 carrying the avirulent effector avrRpt2 [36]. Notably, our investigations unveiled an intriguing facet: OSUB18’s capacity to evoke robust induced systemic resistance (ISR) against *Pst* DC3000 in distal leaves following local immunization via infiltration, both in *Arabidopsis* (Figure 5A,B) and tomato plants (cultivar Heinz) (Figure 5C). Unlike avrRpt2, OSUB18 did not evoke hypersensitive response (HR) symptoms (Figure 5D). This distinctive attribute extended to *Arabidopsis* Col-0 and tomato plants, where OSUB18 likewise stimulated augmented resistance against *B. cinerea* in remote leaves following local immunization via infiltration (Figure 5E,F).

Intrigued by this phenomenon, we postulated the involvement of mobile signals engendered by OSUB18 infiltration, potentially translocated from locally immunized leaves through the plant’s phloem vascular system, thereby culminating in potentiated resistance against bacterial and fungal phytopathogens in distant leaves. To scrutinize this premise, we meticulously harvested phloem exudates (PEXs) from locally immunized leaves by OSUB18, subsequently subjecting these exudates to validation in terms of their potency in eliciting heightened host plant resistance against *Pst* DC3000, both locally (Figure 5G) and systemically (Figure 5H). Our inquiry into the conserved nature of this mechanism led us to explore various other plant growth-promoting rhizobacteria (PGPR) candidates, with outcomes consistently mirroring a parallel mode of action, culminating in the induction of the defense mechanism similar to SAR against both bacterial phytopathogen *Pst* DC3000 (Figure 5I) and fungal phytopathogen *B. cinerea* (Figure 5J).

In summation, our empirical findings provide a compelling elucidation: OSUB18’s inducement of SAR and ISR in host plants stands validated against the dual onslaught of bacterial phytopathogen *Pst* DC3000 and fungal phytopathogen *B. cinerea* [4]. This milestone insight underscores the multifaceted interplay between OSUB18 and the host plant, delineating a promising avenue for advancing our comprehension of plant–microbe interactions in the context of pathogen resistance and defense.

### 3.6. OSUB18 Exhibits Antagonism towards Phytopathogenic Bacteria, Fungi, and Oomycetes

In an antecedent study, we substantiated OSUB18’s capacity to antagonize the bacterial phytopathogen *P. syringae* [37] and the fungal pathogen *B. cinerea* [4], thereby initiating induced systemic resistance. Notwithstanding, the comprehensive extent of OSUB18’s antagonistic efficacy in diverse phytopathogens remains an area of limited elucidation. To expound the breadth of OSUB18’s activity in thwarting phytopathogens, we embarked on an array of OSUB18–phytopathogen co-culture assays employing agar plates as the experimental milieu (Figure 6A). Our rigorous investigations unveiled the striking capacity of OSUB18 to effectively impede the growth of an expansive spectrum of phytopathogens (Figure 6A). This encompassed notable fungal phytopathogens, namely, *Fusarium oxysporum* (Figure 6B) [38,39], *Fusarium graminearum* (Figure 6C) [40,41], and *Magnaporthe oryzae* (Figure 6D) [42,43], alongside oomycete phytopathogens, specifically *Pythium ultimum* (Figure 6E) [44,45] and *Phytophthora capsica* (Figure 6F) [46,47]. The quantification of pathogen inhibition rates revealed a dynamic range spanning from approximately 20% to 50% (Figure 6G). These empirical revelations affirm OSUB18’s remarkable potential as a versatile antagonist against a diverse array of phytopathogens. In summation, our findings concretize OSUB18’s role as a potent deterrent to a spectrum of phytopathogens, thus fortifying its potential utility as a pivotal component in phytopathogen management strategies.

## 4. Discussion

OSUB18, a distinct strain in the *B. proteolyticus* lineage, was identified for its ability to induce ISR against bacterial and fungal pathogens in *Arabidopsis*, as elucidated in our previous study [4]. However, the comprehensive role of OSUB18 in promoting plant growth, enhancing plant tolerance against abiotic stresses, and its intricate modulation of host gene expression remain areas warranting exploration. In this study, our investigation commenced by cultivating *Arabidopsis* Col-0 plants from seeds on sterile agar plates. Employing a controlled approach, we co-cultured these seeds with OSUB18 cells on an MS media plate, ensuring no direct contact, while employing water as a control, as previously mentioned. Our observations unveiled that OSUB18 effectively reshaped root architecture and significantly stimulated seedling growth in *Arabidopsis* Col-0 plants, particularly on a medium consisting of sterile 0.5x MS media (Figure 1A). The consistency of these outcomes was reinforced by the findings from our experimentation with *Arabidopsis* Col-0 plants cultivated in soil, where OSUB18 drenching was consistently associated with growth enhancement [4]. The capacity of OSUB18 to ameliorate plant fitness under varying unfavorable growth conditions, such as cold, salinity, and drought stress, was particularly noteworthy (Figure 1J–M). These findings underscore OSUB18’ s potential to not only foster plant growth in normative conditions, but also in adverse scenarios, possibly attributed to its influence on the root development of host plants.

Evidently, OSUB18 emerges as a promising candidate for application in agricultural systems, offering the prospect of enhanced crop growth and resilience. Prior research has demonstrated the utility of *Bacillus* in safeguarding crops against pests and bolstering yields [48], while scant attention has been devoted to its direct impact on plant growth and the suppression of phytopathogens. Our preceding investigations revealed OSUB18’s origin in switchgrass, unveiling its role in increasing plant fitness and engaging host plant defense mechanisms against the bacterial pathogen *Pst* DC3000 and fungal agent *B. cinerea* [4]. Extending our study’s purview, we delved into the realm of host gene expression patterns via meticulous RNA-seq (Figure 2) and Metascape (Figure 3 and Figure 4) analyses. This exploration unearthed the significant upregulation of glucosinolate biosynthesis and defense-related genes in *Arabidopsis* due to OSUB18 exposure (Figure 3), thus reinforcing its influence on augmenting host plant immunity, as corroborated by our previous study [4]. Conversely, OSUB18-drenching treatment manifested a concurrent downregulation of plant flavonoid biosynthetic and wound responsive genes in *Arabidopsis* (Figure 4). This intriguing finding potentially implicates OSUB18 in facilitating its own colonization in the root rhizosphere of host plants. To augment our understanding, future endeavors may encompass tracking OSUB18’s colonization in host plants. This could be achieved through methods, such as genetically tagging OSUB18 with GFP to monitor its in vivo movement, or through the genomic modification of OSUB18, followed by the design of specific primers for the quantitative assessment of its presence in root and leaf tissues. By addressing these intricacies, we inch closer to a comprehensive understanding of OSUB18’s multifaceted interactions in the areas of plant growth promotion, defense activation, and colonization dynamics.

The well-established phenomenon of beneficial bacteria stimulating host plant immunity against bacterial and fungal pathogens through ISR has been extensively documented [4,35,49,50]. Essentially, the application of beneficial bacteria to host plants via drenching treatment elicits a swifter and more robust defense response to phytopathogen attacks above-ground. However, an intriguing question remains: can a comparable enhancement of host plant immunity be achieved through the immunization of local plant leaves with beneficial bacteria, a mechanism akin to systemic acquired resistance (SAR)? This study seeks to address this inquiry. Contrary to the conventional SAR triggered by pathogenic bacteria, such as the *Pst* DC3000 strain containing avrRpt2 that induces evident HR symptoms in infiltrated local leaves, our investigation of plant leaves infiltrated with OSUB18 unveiled a conspicuous absence of HR symptoms (Figure 5D). Remarkably, the OSUB18 infiltration of local leaves exerted a notable influence on enhancing the defense responses of systemic leaves against both bacterial and fungal pathogens, a phenomenon observed not only in *Arabidopsis*, but also in tomato plants (Figure 5E,F). This observation implies that OSUB18, while inducing an SAR-like effect, does so without eliciting side effects, like HR symptoms, in the host plants. Intriguingly, this SAR-like response is not exclusive to OSUB18, but is also manifest in various other beneficial bacterial strains, such as Pf5 (Figure 5I,J). This suggests that beneficial bacteria may share similar activation pathways with pathogenic bacteria when instigating the SAR mechanism in host plants.

Our endeavor extended to the collection of phloem exudates (PEXs) from OSUB18-infiltrated leaves. The subsequent validation of PEX-OSUB18 activity in inducing SAR against *Pst* DC3000 in *Arabidopsis* plants further affirmed the multifaceted influence of OSUB18 (Figure 5I,J). Yet, a definitive understanding of the precise active molecules in Pex-OSUB18 remains elusive. It is plausible that OSUB18 might stimulate the production of novel signal molecules in host plants, subsequently translocating systemically to bolster resistance in untreated leaf tissues. This hypothesis was bolstered by our PEX treatment results (Figure 5G,H). The identification and characterization of these potential functional signal molecules from the PEX samples of beneficial bacteria warrant further dedicated research. In our study, it was essential to acknowledge some limitations. While our investigations encompassed bacterial and fungal phytopathogens, we did not probe the effects of beneficial bacteria on other types of phytopathogens, such as viruses and herbivores. Therefore, we remained cautious about extending the concept of beneficial bacteria-induced SAR to these unexplored aspects. Further explorations in these directions are essential for a comprehensive understanding of the extent of and mechanisms underlying beneficial bacteria’s impacts on host plant defense mechanisms. This study delved into the intricate interplay between beneficial bacteria and host plants’ defense mechanisms, unveiling novel insights into the mechanisms of SAR and its potential applications in plant protection strategies.

Furthermore, OSUB18 exhibited a notable antagonistic effect against a variety of phytopathogens, encompassing oomycete pathogens, in vitro (Figure 6A). This comprehensive antagonistic activity positioned OSUB18 as a compelling contender for biological control strategies, offering a broad-spectrum efficacy against bacterial, fungal, and oomycete phytopathogens [4]. This study significantly enriches our understanding of the multifaceted potential harbored by the microorganism OSUB18 in combatting destructive plant diseases. Nonetheless, there remains a need for further comprehensive investigations to corroborate OSUB18’s antagonistic efficacy in vivo and against other potential assailants, such as viruses or herbivores. Equally pivotal is the validation of OSUB18’s performance under authentic field conditions, diverging from the controlled settings of a laboratory environment. The natural compounds produced by *Bacillus* spp. display commendable traits of weak environmental persistence and low mammalian toxicity [51]. In this light, *Bacillus* spp. isolates present a promising alternative to synthetic chemicals that pose risks to both living organisms and the environment. Our exploration of the underlying mechanisms through which OSUB18 augments plant fitness in the face of destructive bacterial, fungal, and oomycete phytopathogens augments the arsenal of biological control strategies that foster sustainable agricultural practices and consequently contribute to human health. These findings offer novel perspectives on the practical implementation of OSUB18 as an environmentally friendly biological control alternative, superseding less ecologically sound methods.

Nevertheless, it is imperative to acknowledge certain limitations existing in our study. Firstly, a gap persists in terms of the genetic evidence elucidating OSUB18’s role in triggering ISR or SAR. Addressing this can involve conducting experiments with mutant plants impaired in specific defense signaling pathways, thereby providing a more comprehensive understanding of OSUB18’s mechanisms. Furthermore, the constraints of our study extended to the laboratory-centric nature of our data, warranting a future trajectory toward investigating the dynamics of field conditions. Therefore, validating OSUB18’s efficacy in promoting plant growth and repressing phytopathogens in dynamic environments beyond a controlled laboratory setting is pivotal. Moreover, numerous unknowns persist in the context of ISR or SAR induced by OSUB18. Among these, a pivotal query pertains to the identity of the active molecules in PEX-OSUB18. Additionally, elucidating the mechanisms underpinning the translocation of these molecules from local tissues to distant ones remains an enigma. Are these translocations passive or reliant upon auxiliary proteins?

## 5. Conclusions

The concept of the “disease triangle,” conceived by Dr. George McNew in the 1960s, serves as a framework to comprehend the interplay among various factors contributing to epidemics and to predict, limit, or control these outbreaks [52]. Significantly, this framework notably omits the requisite acknowledgment of the pivotal roles that beneficial microbes assume in the context of the host–pathogen–environment paradigm [37]. This issue bears significance because biological controls possess inherent flexibility that aligns with integrated disease management approaches, effectively preempting potential disease outbreaks from their nascent stages. The incorporation of beneficial microbes into the disease triangle framework thus emerges as a compelling avenue for research and practical applications [37]. The effectiveness of biological control is often compromised when pathogen levels have already escalated. In essence, combatting an ongoing pathogen infestation becomes increasingly challenging as pathogen populations burgeon. The efficacy of biological control is most pronounced when employed as a preventative measure, as highlighted in a previous study [53]. Consequently, harnessing biological control strategies as a preventive approach emerges as a prudent strategy for safeguarding crop yields from phytopathogen contamination during storage and transportation, periods characterized by typically low pathogen levels [37]. By virtue of our findings, we propose an innovative model for beneficial microbe (OSUB18)-mediated plant protection (Figure 7).

Our study aims to introduce a meaningful model that emphasizes the strategic use of beneficial microorganisms, such as OSUB18, to improve and strengthen plant health and yield, addressing the urgent need for innovative and sustainable solutions in modern agriculture. Our study is underpinned by a comprehensive repertoire of treatment methodologies, spanning drenching, spraying, infiltrating treatment, and judicious genetic engineering (of beneficial microbes, such as OSUB18). This multifaceted strategy reveals a panorama of tangible benefits, prominently accentuating amplified plant growth and health across a spectrum of dimensions encompassing root morphology, shoot proliferation, seed yield, and fruit production.

Moreover, our study reveals a remarkable capacity to serve as an adept abiotic stress alleviator, adeptly addressing an array of challenges engendered by nutritional inadequacies, temperature oscillations, salinity-mediated perturbations, and drought-induced constraints. In parallel, the innovative contours of this approach come to the fore as an indispensable biotic stress management tool, efficaciously curbing the pernicious influence of a diverse array of pathogens, spanning viruses, bacteria, fungi, oomycetes, and herbivorous entities.

Beyond these foundational merits, the envisioned model substantiates its potency through the augmentation of the plant’s immune competence, heralding a triad of pivotal defense mechanisms: induced systemic resistance (ISR), systemic acquired resistance (SAR), and herbivore-induced resistance (HIR). In summation, this proposed model stands as an eloquent testament to the symphonic choreography between beneficial microbes (such as OSUB18, LKL04 [54], RRD69 [55], and GD4a [56]) and the host plants (such as *Arabidopsis* [22], tomato [8,9], and strawberry [57] plants), ushering in a transformative epoch in agriculture characterized by heightened yields, fortified resilience against adversities, and the cultivation of sustainable agricultural practices.

In summary, this research underlines the potential use of OSUB18 as a versatile and eco-friendly agent for safeguarding plants against a spectrum of pathogens and diverse abiotic stressors, thereby advancing the prospects of sustainable agriculture while minimizing adverse ecological impacts.

## Figures and Tables

**Figure 1 biology-12-01495-f001:**
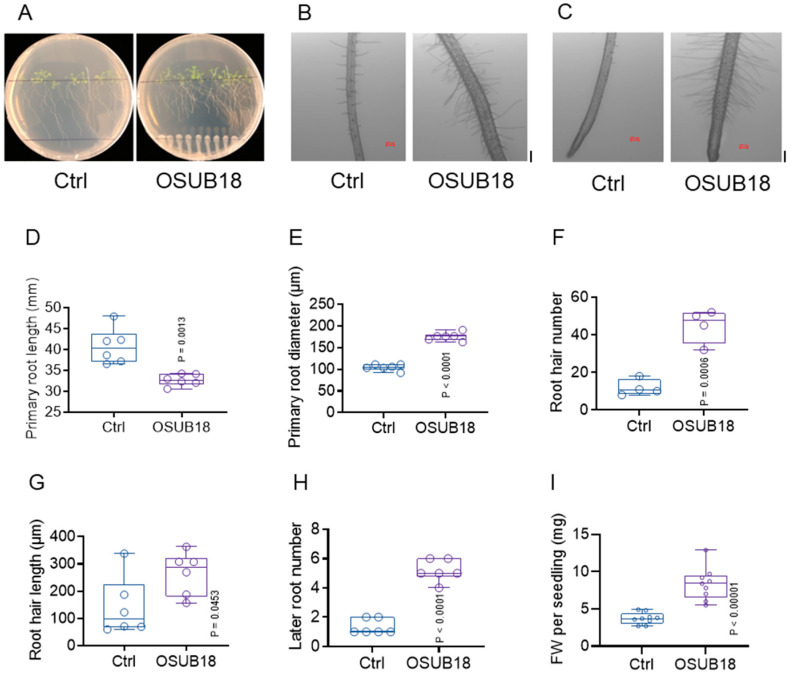
OSUB18 stimulates *Arabidopsis* plants’ later root formation and protects *Arabidopsis* plants from different abiotic stresses. (**A**) OSUB18 inoculation stimulates the lateral root formation of *Arabidopsis* Col-0 plants. (**B**,**C**) OSUB18 inoculation stimulates *Arabidopsis* Col-0 root hair formation. (**B**) Non-tip section; (**C**) root tip section. Scale bar: 0.1 mm. (**D**–**H**) Quantification of the relative primary root length (**D**), primary root size/diameter (**E**), root hair number (**F**), root hair length (**G**), later root number (**H**), and fresh weight (**I**) from (**A**–**C**). (**J**) Representative *Arabidopsis* Col-0 plants grown at room temperature (RT) after treatment with water (Ctrl) or OSUB18. The plant biomass was quantified as FW per shoot (g). Scale bar: 2 cm. (**K**) Representative *Arabidopsis* Col-0 plants grown at 4 °C for one week after the pretreatment with water (Ctrl) or OSUB18. The plant biomass was quantified as FW per shoot (g). Scale bar: 2 cm. (**L**) Representative Col-0 plants grown in 250 mM of NaCl after being drenched with water (Ctrl) or OSUB18. The plant biomass was quantified as FW per shoot (g). Scale bar: 2 cm. (**M**) Representative *Arabidopsis* Col-0 plants grown under drought stress condition after the pretreatment with water (Ctrl) or OSUB18. Upper panels show plants after drought treatment while bottom panels show plants after follow-up rewatering treatment. The plant biomass was quantified as FW per shoot (g). Scale bar: 2 cm.

**Figure 2 biology-12-01495-f002:**
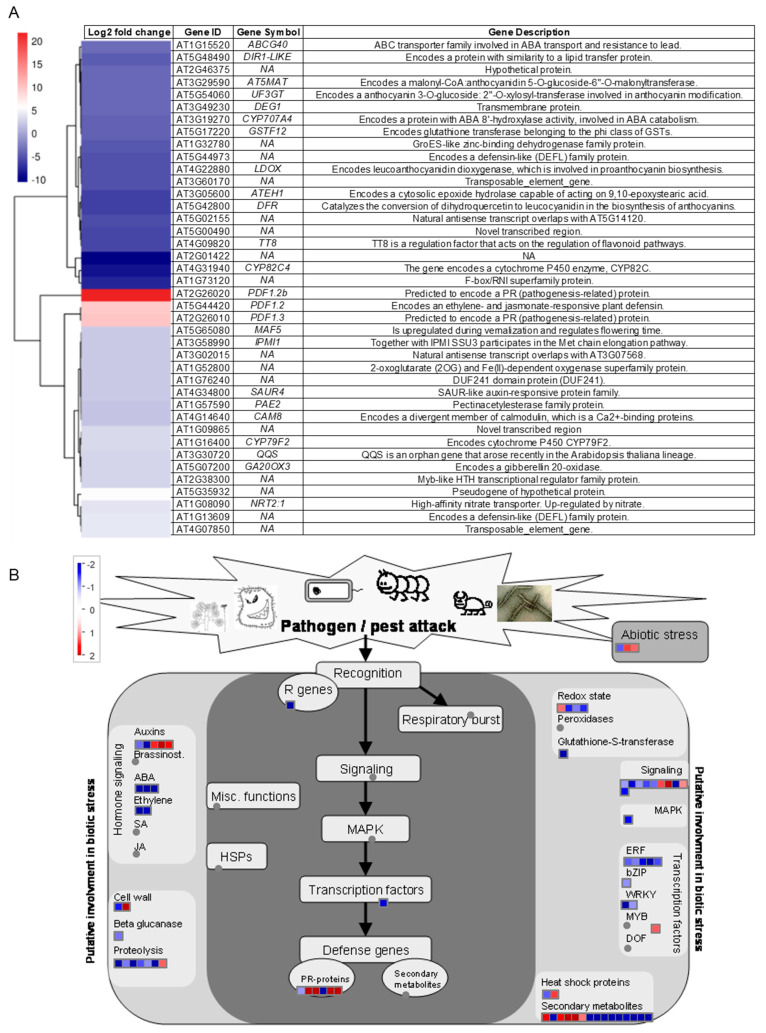
Differentially expressed genes in *Arabidopsis* plants due to OSUB18-drenching treatment. (**A**) Hierarchical clustering of 40 highest or lowest differentially expressed genes due to OSUB18 drenching. The color bar represents the log2 fold change. The thresholds were set as |log2 fold change| > 1 and adjusted *p*-value < 0.05. (**B**) Pathway visualization of genes differentially expressed because of OSUB18-drenching treatment using Mapman annotation software (Version 3.6.0RC1) (|log2 fold change| > 1 and *p*-value adjusted <0.05, OSUB18 drenching versus water control). The color codes represent log2 ratios on the scale bar (red, upregulated; blue, downregulated) corresponding to a log2 ratio of 1. ABA, abscisic acid; SA, salicylic acid; JA, jasmonic acid; R, resistance; MAPK, mitogen-activated protein kinase; PR, pathogenesis-related; ERF, ethylene-responsive factor; bZIP, basic region leucine zipper; MYB, myeloblast; and DOF, DNA-binding with one finger.

**Figure 3 biology-12-01495-f003:**
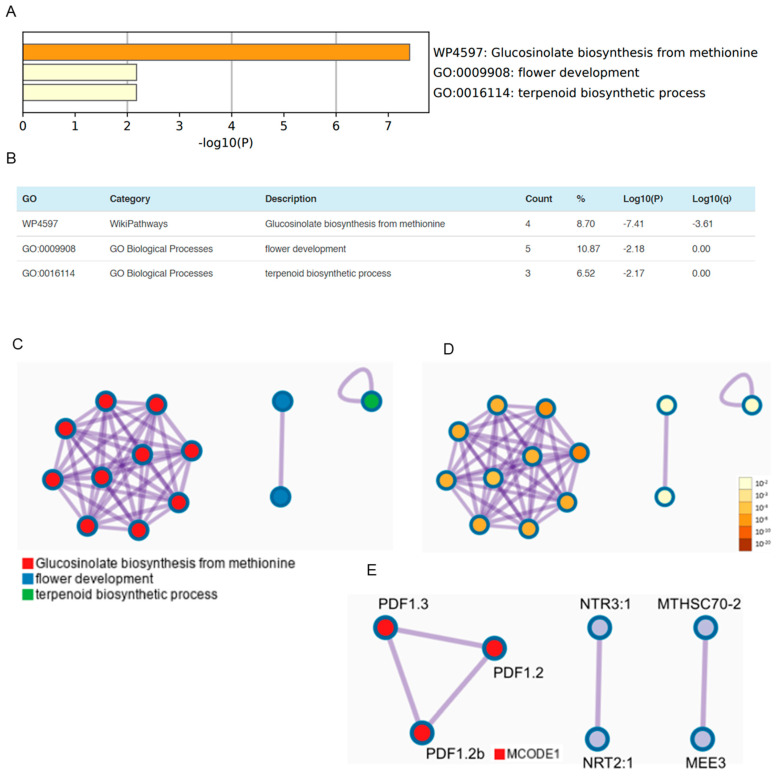
Metascape gene list analysis of the upregulated genes of *Arabidopsis* plants due to OSUB18-drenching treatment. (**A**) Top-level gene ontology (GO) biological processes enriched by OSUB18 drenching, colored by the *p*-value. (**B**) The top-3 GO clusters with their representative enriched terms. “Count” is the number of genes with membership in the given ontology term. “%” is the percentage of all the genes that are found in the given ontology term. “Log10(P)” is the *p*-value in log base 10. “Log10(q)” is the multi-test adjusted *p*-value in log base 10. (**C**) Network of enriched terms colored by the cluster ID, where nodes that share the same cluster IDs are closer to each other. (**D**) Network of enriched terms colored by *p*-value, where terms containing more genes have a more significant *p*-value. (**E**) The protein–protein interaction (PPI) networks identified in the genes upregulated by OSUB18-drenching treatment. Plant defensin 1.2 (*PDF1.2*, *AT5G44420*); plant defensin 1.2b (*PDF1.2b*, *AT2G26020*); plant defensin 1.3 (*PDF1.3*, *AT2G26010*); nitrate transporter 3.1 (*NRT3.1*, *AT5G50200*); nitrate transporter 2.1 (*NRT2:1 AT1G08090*); mitochondrial hso70-2 (*MTHSC70-2*, *AT5G09590*); and maternal effect embryo arrest 3 (*MEE3*, *AT2G21650*).

**Figure 4 biology-12-01495-f004:**
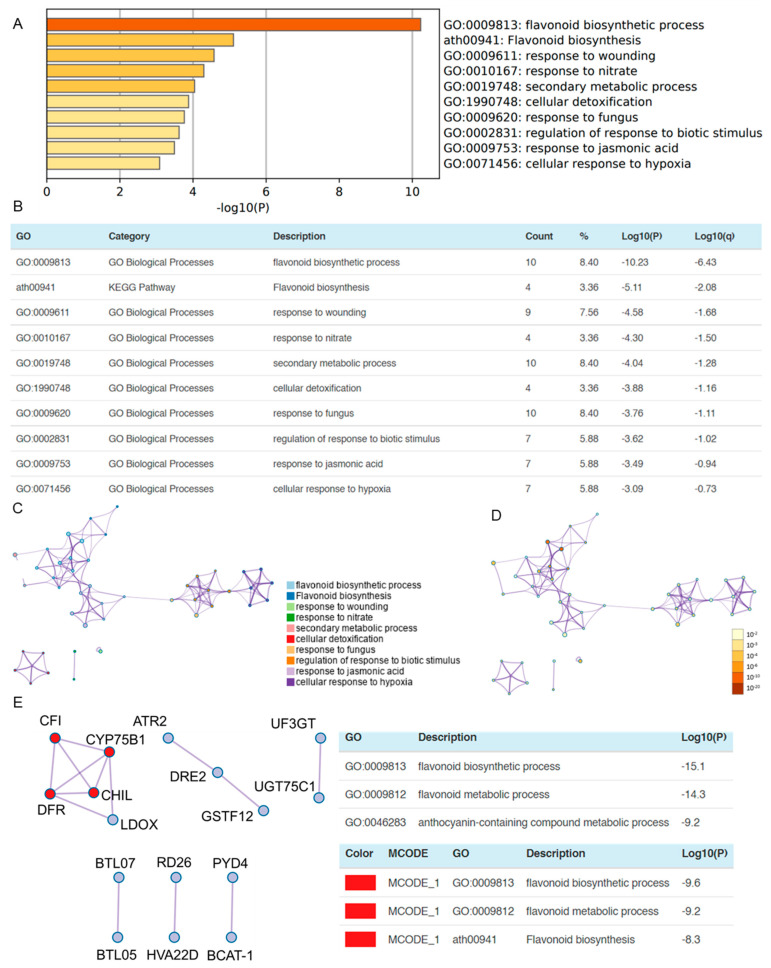
Metascape gene list analysis of the downregulated genes of *Arabidopsis* plants due to OSUB18 drenching. (**A**) Top-level gene ontology (GO) biological processes suppressed by OSUB18 drenching, colored by the *p*-value. (**B**) The top-10 GO clusters with their representative enriched terms. “Count” is the number of genes with a membership in the given ontology term. “%” is the percentage of all the genes that are found in the given ontology term. “Log10(P)” is the *p*-value in log base 10. “Log10(q)” is the multi-test adjusted *p*-value in log base 10. (**C**) Network of enriched terms colored by cluster ID, where nodes that share the same cluster IDs are closer to each other. (**D**) Network of enriched terms colored by *p*-value, where terms containing more genes have a more significant *p*-value. (**E**) The protein–protein interaction (PPI) network and molecular complex detection (MCODE) component identified in the genes downregulated by OSUB18 drenching. Chalcone flavanone isomerase (*CFI*, *AT3G55120*); cytochrome P450 75B1 (*CYP75B1*, *AT5G07990*); dihydroflavonol 4-reductase (*DFR*, *AT5G42800*); chalcone isomerase like (*CHIL*, *AT5G05270*); leucoanthocyanidin dioxygenase (*LDOX*, *AT4G22880*); P450 reductase 2 (*ATR2*, *AT4G30210*); homolog of yeast dre2 (*DRE2*, *AT5G18400*); glutathione s-transferase phi 12 (*GSTF12*, *AT5G17220*); UDP-glucose:flavonoid 3-o-glucosyltransferase (*UF3GT*, *AT5G54060*); UDP-glycosyltransferase 75C1 (*UGT75C1*, *AT4G14090*); zinc finger atl 7 (*BTL07*, *AT3G13430*); zinc finger atl 5 (*BTL05*, *AT4G26400*); responsive to desiccation 26 (*RD26*, *AT4G27410*); hva22 homolog D (*HVA22D*, *AT4G24960*); pyrimidine 4 (*PYD4*, *AT3G08860*); and branched-chain amino acid transaminase 1 (*BCAT1*, *AT1G10060*).

**Figure 5 biology-12-01495-f005:**
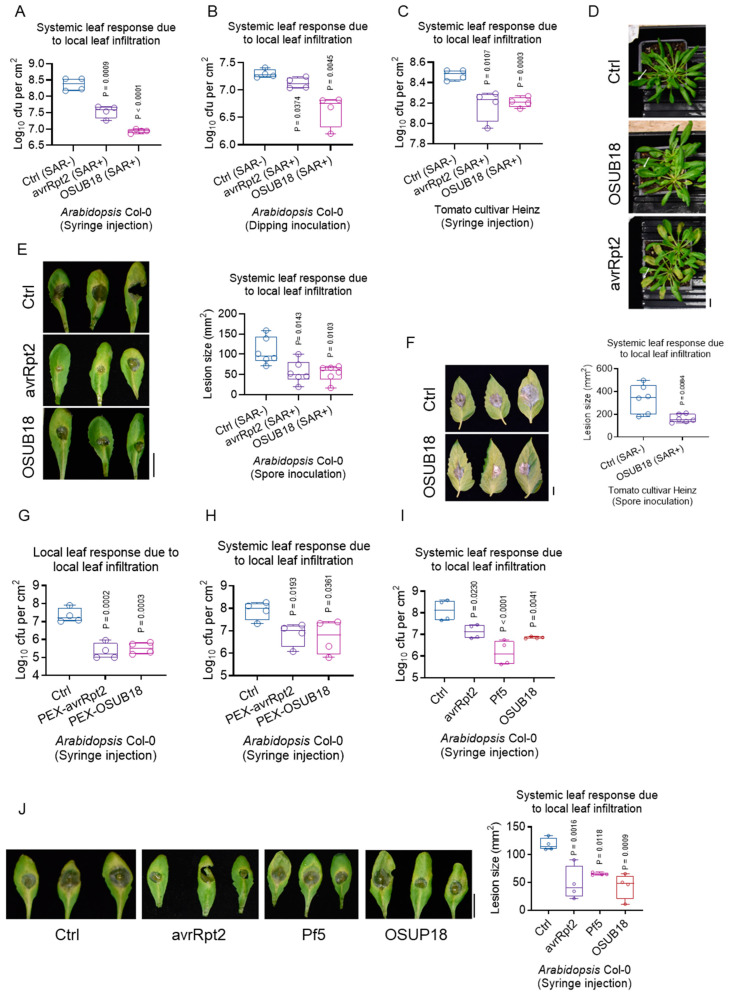
OSUB18 induces the systemic immunity of *Arabidopsis* plants after local leaf infiltration. (**A**,**B**) OSUB18 local infiltration activates the SAR of *Arabidopsis* Col-0 plants against the bacterial phytopathogen *P. syringae*. A syringe injection of *Pst* DC3000 was performed in (**A**), while the dipping inoculation of *Pst* DC3000 was performed in (**B**). avrRpt2 was used as the positive control for SAR induction (SAR+). Water was used as the negative control for SAR induction (SAR−). (**C**) OSUB18 local infiltration activates the SAR of tomato plants against the bacterial phytopathogen *P. syringae*. avrRpt2 was used as the positive control for SAR induction (SAR+). Water was used as the negative control for SAR induction (SAR−). (**D**) OSUB18 is not opportunistically pathogenic to *Arabidopsis* Col-0 plants. Representative 2dpi Col-0 plants infiltrated with water (up panel), OSUB18 (middle panel), or avrRpt2 (bottom panel) are shown. Representative infiltrated leaves are indicated by the white arrows. Note there is no disease symptom or hypersensitive cell death response (HR) in the upper (water infiltration) or middle (OSUB18 infiltration) panels, while HR is obvious in the bottom (avrRpt2 infiltration) panel. Scale bar: 1 cm. (**E**) OSUB18 local infiltration activates the SAR of *Arabidopsis* Col-0 plants against the fungal pathogen *B. cinerea*. Scale bar: 1 cm. avrRpt2 was used as the positive control for SAR induction (SAR+). Water was used as the negative control for SAR induction (SAR−). (**F**) OSUB18 local infiltration activates the SAR of tomato plants against the fungal pathogen *B. cinerea*. Scale bar: 1 cm. avrRpt2 was used as the positive control for SAR induction (SAR+). Water was used as the negative control for SAR induction (SAR−). (**G**) PEX-OSUB18 induces the local plant defense of *Arabidopsis* Col-0 plants against *Pst* DC3000. (**H**) PEX-OSUB18 induces the systemic plant defense of *Arabidopsis* Col-0 plants against *Pst* DC3000. (**I**) Beneficial bacteria induce the systemic plant defense of *Arabidopsis* Col-0 plants against *Pst* DC3000. (**J**) Beneficial bacteria induce the systemic plant defense of *Arabidopsis* Col-0 plants against *B. cinerea*. Scale bar: 1 cm.

**Figure 6 biology-12-01495-f006:**
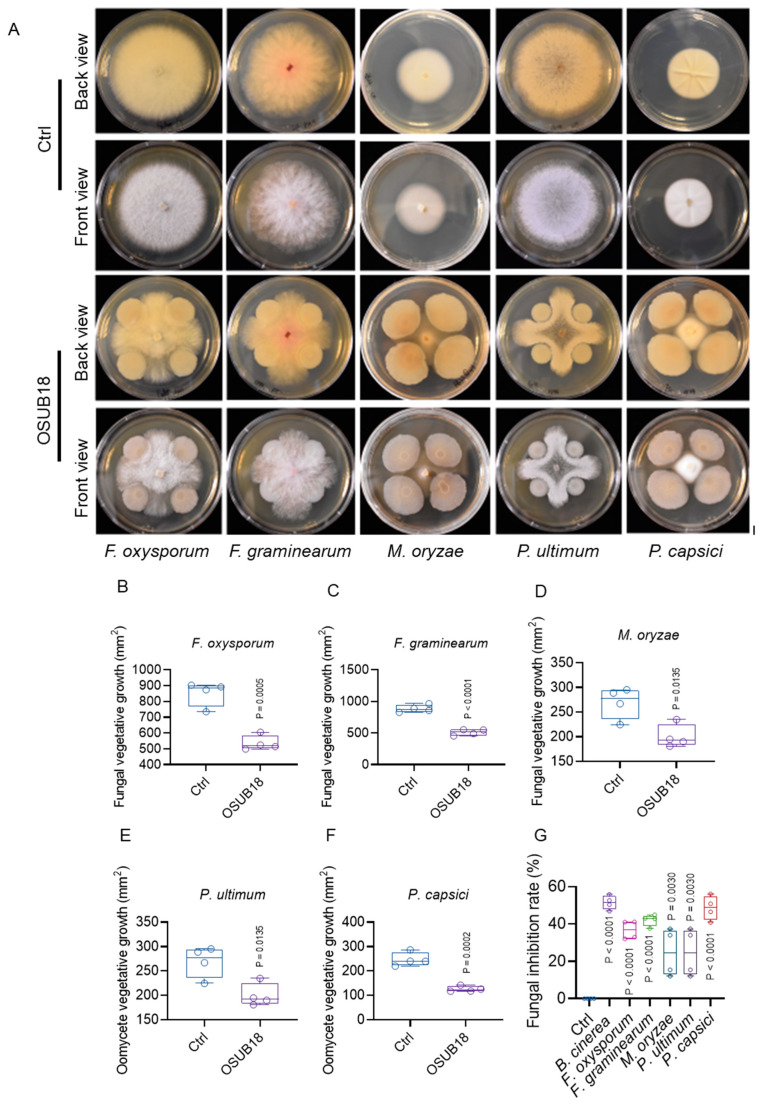
OSUB18 antagonizes multiple phytopathogenic fungi and oomycetes. (**A**) Growth of different phytopathogens on agar plates co-cultured with water (Ctrl) or OSUB18. Scale bar: 1 cm. (**B**) Quantification of the growth of *Fusarium oxysporum* in (**A**). (**C**) Quantification of the growth of *F. graminearum* in (**A**). (**D**) Quantification of the growth of *Magnaporthe oryzae* in (**A**). (**E**) Quantification of the growth of *Pythium ultimum* in (**A**). (**F**) Quantification of the growth of *Phytophthora capsici* in (**A**). (**G**) Quantification of the inhibition rate of fungal or oomycete phytopathogens by OSUB18 in (**A**).

**Figure 7 biology-12-01495-f007:**
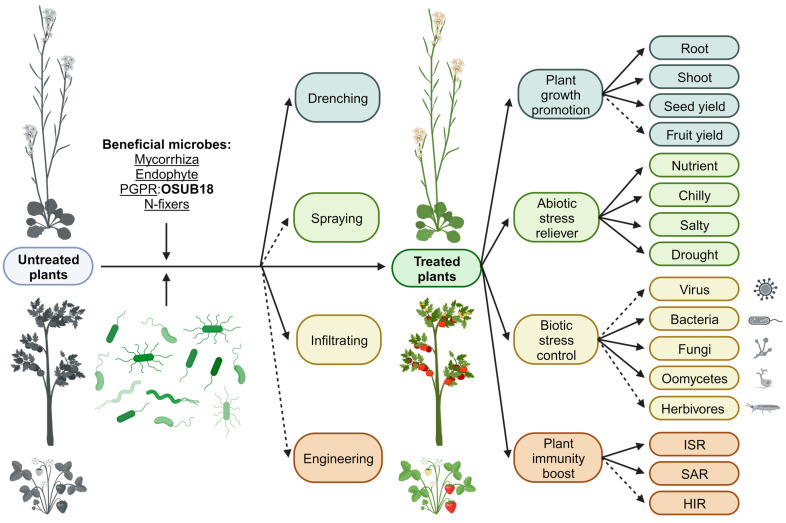
A proposed model of beneficial microbe-mediated plant protection. The proposed model of beneficial microbe-mediated plant protection presents a comprehensive approach to enhancing agricultural sustainability. Employing diverse treatment methods, such as drenching, spraying, infiltration, and genetic engineering, this innovative strategy offers multifaceted benefits. It excels in promoting plant growth across various aspects, including root and shoot development, and seed and fruit yield augmentation. Additionally, this model acts as a remarkable abiotic stress alleviator, mitigating the challenges presented by nutrient deficiencies, extreme temperatures, salinity, and drought conditions. Simultaneously, it demonstrates remarkable efficacy in biotic stress management, efficiently controlling threats from viruses, bacteria, fungi, oomycetes, and herbivores. Moreover, the proposed model significantly elevates plant immunity through induced systemic resistance (ISR), systemic acquired resistance (SAR), and heightened induced resistance (HIR), resulting in a fortified and resilient plant defense system. This integrated approach holds immense promise for revolutionizing agricultural practices and ensuring global food security. This model was created with BioRender.com and is not drawn to scale for clarity. The solid arrow indicates the availability of direct supporting data from studying OSUB18. The dashed arrow indicates potential possibilities not yet tested on OSUB18.

## Data Availability

The data presented in this study are available in the article. Further information is available upon request from the corresponding author.

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
