# Peer review of "Plant Growth Promotion and Stress Tolerance Enhancement through Inoculation with Bacillus proteolyticus OSUB18"

_biology, 2023, doi:10.3390/biology12121495_

Round 1

Reviewer 1 Report

Comments and Suggestions for Authors

Dear Authors!

The article is devoted to the study of the effect of B. proteolyticus strain OSUB18 on the model plant Arabidopsis thaliana. It is shown that treatment of the plant with OSUB18 strain has a positive effect on growth, root system architecture, and resistance to biotic and abiotic stresses. Using RNA-seq technology, many differentially expressed genes relevant to the interaction between A. thaliana and OSUB18, as well as genes relevant to stress tolerance, were identified. OSUB18 has been shown to enhance resistance against phytopathogens, bacteria, fungi and oomycetes. The work contains significant new information of both theoretical interest and possible future practical importance.

The work is large, carefully done, well illustrated and I have no serious remarks in my field of expertise.

However, I would like to draw the attention of the authors to some elements.

Lines 134, 148. The authors for some reason grow A. thaliana under short day conditions (8 hours light) and in another case a 12 hour light period. Why not 16 h light?

 Lines-155-157 A more detailed description of the conditions of NaCl treatment of plants and the creation of drought would be highly desirable. From the authors' description it is impossible to understand the exact set-up of the experiment.

 Lines 240 - 243. "However, insights into OSUB18's contributions to plant root development and related function have not been well studied. Leveraging the premise of OSUB18's capacity to yield beneficial metabolites [4], we postulate a potential transformation in the root architecture of Arabidopsis Col-0 plants post OSUB18 treatment."

This phrase seems unfortunate for two reasons. Firstly, if the authors are talking about the release of some important metabolites by bacteria, it would be better to say here what kind of metabolites. Also, the secretion of metabolites does not seem to me to be sufficient to suggest a change in the architecture of the plant's roots. It is more likely that the authors first saw the change in root architecture and then speculated.

 Lines 260-263 "These collective findings duly validate our underlying hypothesis that OSUB18 exerts a growth-promoting influence in stress conditions, potentially mediated through a reconfiguration of root architecture within the context of Arabidopsis Col-0 plants."

 It seems to me that the authors overemphasise root architecture and try to explain almost everything by changes in root architecture. The root has become shorter and thicker and what of it? How might this affect growth and especially stress tolerance.

 Lines 342-344 The authors found that OSUB18 cause significant activation of expression of three groups of genes, including genes involved in flower formation, but this group of genes is not discussed in any way in the paper. Why can this be so and how is it beneficial to the plant?

 It is far from clear what has been done before and what is done for the first time in this article.

Reviewer 2 Report

Comments and Suggestions for Authors

This study investigates plant growth promotion and stress tolerance enhancement through inoculation with B. proteolyticus OSUB18. Several parts in the material and method were not clear, In 2.1, authors said the arabidopsis or tomato plants were grown in soil, what kind of soil? in pot? how big of the pot? How did they culture the plants? How many plants?

Major comments:

In the abstract, the authors did not show the main findings, which need to be rewritten.

In 2.2, they said the Arabidopsis seedlings were drenched with OSUB18 with 107CFU/ml, did they check what the bacteria does after the drenching?

In 2.3, For RNA-seq analysis, how many replicates did they use? Arabidopsis plants after three times drenching treatments (OSUB18 VS water), what is the time interval between the treatments? Did the author use the soil?

The authors need to introduce avrRpt2 in the material and method.

In your previous paper, Yang et al., 2023 Frontiers in Plant Science, Fig5A and 5B, which were quite similar to Fig 5 E and 5F in this manuscript, but the lesion size is quite different, why? the authors need to introduce which fungi used in Fig 5E.

What is the mechanism for the OSUB18 exhibiting antagonism towards phytopathogenic bacteria, fungi, and oomycetes? what kinds of antibiotics are produced?

Will OSUB18 show a biological control effect against fungi in a greenhouse test?

Minor comments:

Arabidopsis thaliana in simple summary should be in italics.

Species' names (family, genus, species, and variety or subspecies) should be written in italics.

The line space of paragraph 2.4 should change.

The quality of Fig 2B, Fig 3, and Fig 4 is low.

Comments on the Quality of English Language

No

Round 2

Reviewer 2 Report

Comments and Suggestions for Authors

No more comments.

Author Response

We sincerely thank Reviewer 2 for his/her thoughtful and beneficial feedback on our manuscript. His/her encouraging comments and detailed critiques have greatly improved the quality of our work.